# Meta-Black-Box-Optimization through Offline Q-function Learning

Zeyuan Ma [1]  Zhiguang Cao [2]  Zhou Jiang [1]  Hongshu Guo [1]  Yue-Jiao Gong [1]

## Abstract

Recent progress in Meta-Black-Box-Optimization (MetaBBO) has demonstrated that using RL to learn a meta-level policy for dynamic algorithm configuration (DAC) over an optimization task distribution could significantly enhance the performance of the low-level BBO algorithm. However, the online learning paradigms in existing works makes the efficiency of MetaBBO problematic. To address this, we propose an offline learning-based MetaBBO framework in this paper, termed Q-Mamba, to attain both effectiveness and efficiency in MetaBBO. Specifically, we first transform DAC task into long-sequence decision process. This allows us further introduce an effective Q-function decomposition mechanism to reduce the learning difficulty within the intricate algorithm configuration space. Under this setting, we propose three novel designs to meta-learn DAC policy from offline data: we first propose a novel collection strategy for constructing offline DAC experiences dataset with balanced exploration and exploitation. We then establish a decomposition-based Q-loss that incorporates conservative Q-learning to promote stable offline learning from the offline dataset. To further improve the offline learning efficiency, we equip our work with a Mamba architecture which helps long-sequence learning effectiveness and efficiency by selective state model and hardware-aware parallel scan respectively. Through extensive benchmarking, we observe that Q-Mamba achieves competitive or even superior performance to prior online/offline baselines, while significantly improving the training efficiency of existing online baselines. We provide sourcecodes of Q-Mamba online.

---

[1]South China University of Technology, China [2]Singapore Management University, Singapore. Correspondence to: Yue-Jiao Gong <gongyuejiao@gmail.com>.

*Proceedings of the 42nd International Conference on Machine Learning*, Vancouver, Canada. PMLR 267, 2025. Copyright 2025 by the author(s).

## 1. Introduction

Black-Box-Optimization (BBO) problem is challenging due to agnostic problem formulation, requiring effective BBO algorithms such as Evolutionary Computation (EC) to address (Zhan et al., 2022). For decades, various EC methods have been crafted by experts in optimization domain to solve diverse BBO problems (Slowik & Kwasnicka, 2020; Guo et al., 2024c). However, one particular technical bottleneck of human-crafted BBO algorithms is that they hardly generalize across different problems (Eiben & Smit, 2011). Deep expertise is required to adapt an existing BBO algorithm for novel problems, which impedes EC's further spread.

Recent Meta-Black-Box-Optimization (MetaBBO) works address the aforementioned generalization gap by introducing a bi-level learning to optimize paradigm (Ma et al., 2024d; Yang et al., 2025), where a neural network-based policy (e.g., RL (Sutton, 2018)) is maintained at the meta level and meta-trained to serve as experts for configuring the low-level BBO algorithm to attain maximal performance gain on a problem distribution. Though promising, existing MetaBBO methods suffer from two technical bottlenecks: a) **Learning Effectiveness:** some advanced BBO algorithms hold massive configuration spaces with various hyper-parameters, which is challenging for RL to learn effective DAC policy. **Training Efficiency**: Existing MetaBBO methods employ online RL paradigm, showing inefficiency for sampling training trajectories.

Given a such dilemma in-between the effectiveness and efficiency, we in this paper propose an offline MetaBBO framework, termed Q-Mamba, to meta-learn effective DAC policy through an offline RL pipeline. Specifically, to reduce the difficulty of searching for optimal policy from the entire configuration space of BBO algorithms, we first transform DAC task into a long-sequence decision process and then introduce a Q-function decomposition scheme to represent each hyper-parameter in the BBO algorithm as a single action step in the decision process. This allows us to learn Q-policy for each hyper-parameter in an autoregressive manner. We propose three core designs to support offline RL under such setting: a) **Offline data collection strategy**: We collect offline DAC experience trajectories from both strong MetaBBO baselines and a random policy to provide exploitation and exploration data used for robust training.

b) **Conservative Q-learning**: we proposed a compositional Q-loss that integrates conservative loss term (Kumar et al., 2020) to address endemic distributional shift issue (Wang et al., 2021) in offline RL. c) **Mamba-based RL agent**: the Q-function decomposition scheme would make the DAC decision process in Q-Mamba much longer than those in existing MetaBBO. We hence design a Mamba-based neural network architecture as the RL agent, which shows stronger long-sequence learning capability through its selective state model, and appealing training efficiency through its parallel scan on whole trajectory sample. Accordingly, we summarize our contributions in three-folds:

i) Our main contribution is Q-Mamba, the first offline MetaBBO framework which shows superior learning effectiveness and efficiency to prior MetaBBO baselines.

ii) To ensure offline learning effectiveness, a Q-function decomposition scheme is embedded into the DAC decision process of BBO algorithm which facilitates separate Q-function learning for each action dimension. Besides, a novel data collection strategy constructs demonstration dataset with diversified behaviours, which can be effectively learned by Q-Mamba through a compositional Q-loss which enhances the offline learning by removing distributional shift. To further improve the training efficiency, we design a Mamba-based RL agent which seamlessly aligns with the Q-function decomposition scheme and introduces desirable training acceleration compared to Transformer structures, through parallel scan.

iii) Experimental results show that our Q-Mamba effectively achieves competitive or even superior optimization performance to prior online/offline learning baselines, while consuming at most half training budget of the online baselines. The learned meta-level policy can also be readily applied to enhance the performance of the low-level BBO algorithm on unseen realistic scenarios, e.g., Neuroevolution (Such et al., 2017) on continuous control tasks.

## 2. Related Works

### 2.1. Meta-Black-Box-Optimization

Meta-Black-Box-Optimization (MetaBBO) aims to learn the optimal policy that boosts the optimization performances of the low-level BBO algorithm over an optimization problems distribution (Ma et al., 2024d). Although several works facilitate supervised learning (Chen et al., 2017; Song et al., 2024; Li et al., 2024b;c; Wu et al., 2023), Neuroevolution (Lange et al., 2023b;a; Ma et al., 2024a) or even LLMs (Ma et al., 2024c; Liu et al., 2024) to meta-learn the policy, the majority of current MetaBBO methods adopt reinforcement learning for the policy optimization to strike a balance between effectiveness and efficiency (Li et al., 2024a; Ma et al., 2023). Specifically, the dynamic algorithm

configuration (DAC) during the low-level optimization can be regarded as a Markov Decision Process (MDP), where the state reflects the status of the low-level optimization process, action denotes the configuration space of the low-level algorithm and a reward function is designed to provide feedback to the meta-level control policy. Existing MetaBBO methods differ with each other in the configuration space. In general, the configuration space of the low-level algorithm involves the operator selection and/or the hyper-parameter tuning. For the operator selection, initial works such as DE-DDQN (Sharma et al., 2019) and DE-DQN (Tan & Li, 2021) facilitate Deep Q-network (DQN) (Mnih, 2013) as the meta-level policy and dynamically suggest one of the prepared mutation operators for the low-level Differential Evolution (DE) (Storn & Price, 1997) algorithm. Following such paradigm, PG-DE (Zhang et al., 2024) and RL-DAS (Guo et al., 2024a) further explore the possibility of using Policy Gradient (PG) (Schulman et al., 2017) methods for the operator selection and demonstrate PG methods are more effective than DQN methods. Besides, RLEMMO (Lian et al., 2024) and MRL-MOEA (Wang et al., 2024) extend the target optimization problem domain from single-objective optimization to multi-modal optimization and multi-objective optimization respectively. Unlike the operator selection, the action space in hyper-parameter tuning is not merely discrete since typically the hyper-parameters of an algorithm are continuous with feasible ranges. In such continuous setting, the action space is infinite and can be handled either by discretizing the continuous value range to reduce this space (Liu et al., 2019; Xu & Pi, 2020; Hong et al., 2024; Yu et al., 2024) or directly using PG methods for continuous control (Yin et al., 2021; Sun et al., 2021; Wu & Wang, 2022; Ma et al., 2024b).

While simply doing operator selection or hyper-parameter tuning for part of an algorithm has shown certain performance boost, recent MetaBBO researches indicate that controlling both sides gains more (Xue et al., 2022; Zhao et al., 2024). In particular, an up-to-date work termed as ConfigX (Guo et al., 2024b) constructs a massive algorithm space and has shown possibility of meta-learning a universal configuration agent for diverse algorithm structures. However, the massive action space in such setting and the online RL process in these MetaBBO methods make it challenging to balance the training effectiveness and the efficiency.

### 2.2. Offline Reinforcement Learning

Offline RL (Levine et al., 2020) aims at learning the optimal control policy from a pre-collected demonstration set, without the direct interaction with the environment. This is appealing for real-world complex control tasks, where on-policy data collection is extremely time-consuming (i.e., the dynamic algorithm configuration for black-box optimization discussed in this paper). A critical challenge in

offline RL is the distribution shift (Fujimoto et al., 2019): learning from offline data distribution might mislead the policy optimization for out-of-distribution transitions hence degrades the overall performance. Common practices in offline RL to relieve the distribution shift include a) learning policy model (e.g., Q-value function) by sufficiently exploiting the Bellman backups of the transition data in the demonstration set and constraining the value functions for out-of-distribution ones (Haarnoja et al., 2018; Kumar et al., 2020). b) conditional imitation learning (Chen et al., 2021; Janner et al., 2021; Dai et al., 2024b) which turns the MDP into sequence modeling problem and uses sequence models (e.g., recurrent neural network, Transformer or Mamba) to imitate state-action-reward sequences in the demonstration data. Although the conditional imitation learning methods have been used successfully in control domain, they have stitching issue: they do not provide any mechanism to improve the demonstrated behaviour as those policy model learning methods. To address this, QDT (Yamagata et al., 2023) and QT (Hu et al., 2024) additionally train a value network to relabel the return-to-go in offline dataset, so as to attain stitching capability. Differently, Q-Transformer (Chebotar et al., 2023) combines the strength of both lines of works by first decomposing the Q-value function for the entire high-dimensional action space into separate one-dimension Q-value functions, and then leveraging transformer architecture for sequential Bellman backups learning. Q-Transformer allows policy improvement during the sequence-to-sequence learning hence achieves superior performance to the prior works.

## 3. Preliminaries

### 3.1. Decomposed Q-function Representation

Suppose we have a MDP $\{S, A = (A_1, ..., A_K), R, \mathcal{T}, \gamma\}$, where the action space is associated by a series of $K$ action dimensions, $S$, $R(S, A)$, $\mathcal{T}(S'|S, A)$, $\gamma$ denote the state, reward function, transition dynamic and discount factor, respectively. Value-based RL methods such as Q-learning (Watkins & Dayan, 1992) learn a Q-function $Q(s^t, a_{1:K}^t)$ as the prediction of the accumulated return from the time step $t$ by applying $a_{1:K}^t$ at $s^t$. The Q-function can be iteratively approximated by Bellman backup:

$$Q(a_{1:K}^t|s^t) \leftarrow R(s^t, a_{1:K}^t) + \gamma \max_{a_{1:K}^{t+1}} Q(a_{1:K}^{t+1}|s^{t+1}). \quad (1)$$

However, suppose there are at least $M$ action bins for each of the $K$ action dimensions, the Bellman backup above would be problematic since the associated action space contains $M^K$ feasible actions. Such dimensional curse challenges the learning effectiveness of the value-based RL methods. Recent works such as SDQN (Metz et al., 2017) and Q-Transformer (Chebotar et al., 2023) propose decomposing the associated Q-functions into series of time-

dependent Q-function representations for each action dimension to escape the curse of dimensionality. For the $i$-th action dimension, the decomposed Q-function is written as:

$$Q(a_i^t|s^t) \leftarrow \begin{cases} \max_{a_{i+1}^t} Q(a_{i+1}^t|s^t, a_{1:i}^t), & if \quad i < K \\ R(s^t, a_{1:K}^t) + \gamma \max_{a_1^{t+1}} Q(a_1^{t+1}|s^{t+1}). \\ & if \quad i = K \end{cases}$$

$$(2)$$

Such a decomposition allows using sequence modeling techniques to learn the optimal policy effectively, while holding the learning consistency with the Bellman backup in Eq. (1). We provide a brief proof in Appendix A.

### 3.2. State Space Model and Mamba

For an input sequence $x \in \mathbb{R}^{L \times D}$ with time horizon $L$ and $D$-dimensional signal channels at each time step, State Space Model (SSM) (Gu et al., 2022) processes it by the following first-order differential equation, which maps the input signal $x(t) \in \mathbb{R}^D$ to the time-dependent output $y(t) \in \mathbb{R}^D$ through implicit latent state $h(t)$ as follows:

$$h(t) = \overline{A}h(t-1) + \overline{B}x(t), \quad y(t) = Ch(t). \quad (3)$$

Here, $\overline{A}$, $\overline{B}$ and $C$ are learnable parameters, $\overline{A}$ and $\overline{B}$ are obtained by applying zero-order hold (ZOH) discretization rule. An important property of SSM is linear time invariance. That is, the dynamic parameters (e.g., $\overline{A}$, $\overline{B}$ and $C$) are fixed for all time steps. Such models hold limitations for sequence modeling problem where the dynamic is time-dependent. To address this bottleneck, Mamba (Gu & Dao, 2023) lets the parameters $\overline{B}$ and $C$ be functions of the input $x(t)$. Therefore, the system now supports time-varying sequence modeling. In the rest of this paper, we use mamba_block() to denote a Mamba computation block described in Eq. (3).

## 4. Q-Mamba

### 4.1. Problem Formulation

A MetaBBO task typically involves three key ingredients: a neural network-based meta-level policy $\pi_\theta$, a BBO algorithm $A$ and a BBO problem distribution $P$ to be solved.

**Optimizer $A$.** BBO algorithms such as Evolutionary Algorithms (EAs) have been discussed and developed over decades. Initial EAs such as Differential Evolution (DE) (Storn & Price, 1997) holds few hyperparameters (only two, $F$ and $Cr$ for balancing the mutation and crossover strength). Modern variants of DE integrate various algorithmic components to enhance the optimization performance. Taking the recent winner DE algorithm in *IEEE CEC Numerical Optimization Competition* (Mohamed et al., 2021), MadDE (Biswas et al., 2021) as an example, it

has more than ten hyper-parameters, which take either continuous or discrete values. Hence, the configuration space of MadDE is exponentially larger than original DE. In this paper, we use $A : \{A_1, A_2, ..., A_K\}$ to represent an algorithm with $K$ parameters. We use additional $a_i$ to represent the taken value of $A_i$.

**Problem distribution $P$.** By leveraging the generalization advantage of meta-learning, MetaBBO trains $\pi_\theta$ over a problem distribution $P$. A common choice of $P$ in existing MetaBBO works is the *CoCo BBOB Testsuites* (Hansen et al., 2021), which contains 24 basic synthetic functions, each can be extended to numerous problem instances by randomly rotating and shifting the decision variables. Training on all problem instances in $P$ is impractical. We instead sample a collection of $N$ instances $\{f_1, f_2, ..., f_N\}$ from $P$ as the training set. For the $j$-th problem $f_j$, we use $f_j^*$ to represent its optimal objective value, and $f_j(x)$ as the objective value at solution point $x$.

For an algorithm $A$ and a problem instance $f_j$, suppose we have a control policy $\pi_\theta$ at hand and we use $A$ to optimize $f_j$ for $T$ time steps (generations). At the $t$-th generation, we denote the solution population as $X^t$. An optimization state $s^t$ is first computed to reflect the optimization status information of the current solution population $X^t$ and the corresponding objective values $f_j(X^t)$. Then the control policy dictates a desired configuration for $A$: $a_{1:K}^t = \pi_\theta(s^t)$. $A$ optimizes $X^t$ by $a_{1:K}^t$ and obtains an offspring population $X^{t+1}$. A feedback reward $R(s^t, a_{1:K}^t)$ can then be computed as a measurement of the performance improvement between $f_j(X^t)$ and $f_j(X^{t+1})$. The meta-objective of MetaBBO is to search the optimal policy $\pi_{\theta^*}$ that maximizes the expectation of accumulated performance improvement over all problem instances in the training set:

$$\theta^* = \arg\max_\theta \frac{1}{N} \sum_{j=1}^{N} \sum_{t=1}^{T} R(s^t, a_{1:K}^t | \pi_\theta), \qquad (4)$$

where such a meta-objective can be regarded as MDP. An effective policy search technique for solving MDP is RL, which is widely adopted in existing MetaBBO methods. In this paper, we focus on a particular type of RL: Q-learning, which performs prediction on the Q-function in a dynamic programming way, as described in Eq. (1).

### 4.2. Offline E&E Dataset Collection

The trajectory samples play a key role in offline RL applications (Ball et al., 2023). On the one hand, good quality data helps the training converges. On the other hand, randomly generated data help RL explore and learn more robust model. In Q-Mamba, we collect a trajectory dataset $\mathbb{C}$ of size $D = 10K$ which combines the good quality data and randomly generated data. Concretely, for a low-level BBO algorithm $A$ with $K$ hyper-parameters and a problem dis-

tribution $P$, we pre-train a series of up-to-date MetaBBO methods (e.g., RLPSO (Wu & Wang, 2022), LDE (Sun et al., 2021), GLEET (Ma et al., 2024b)) which control hyper-parameters of $A$ to optimize the problems in $P$. Then we rollout the pre-trained MetaBBO methods on problem instances in $P$ to collect $\mu \cdot D$ complete trajectories. We then use the random strategy to randomly control the hyper-parameters of $A$ to optimize the problems in $P$ and collect $(1 - \mu) \cdot D$ trajectories. By combining the exploitation experience in the trajectories of MetaBBO methods and the exploration experience in the random trajectories, Q-Mamba learns robust and high-performance meta-level policy. In this paper, we set $\mu = 0.5$ to strike a good balance.

### 4.3. Conservative Q-learning Loss

Online learning is widely adopted in existing works, which is especially inefficient under MetaBBO setting, where the low-level optimization typically involves hundreds of optimization steps hence extremely time-consuming. In this paper we propose learning the decomposed sequential Q-function through offline RL to improve the training efficiency of MetaBBO. Concretely, we consider a trajectory $\tau = \{s^1, (a_1^1, ..., a_K^1), r^1, ..., s^T, (a_1^T, ..., a_K^T), r^T\}$, which is previously sampled by an offline policy $\hat{\pi}$. Here, $a_i^t$ denotes the action bin selected for $A_i$ at time step $t$. The training objective of Q-Mamba is a synergy of Bellman backup update (Eq. (2)) and conservative regularization as

$$J(\tau|\theta) = \sum_{t=1}^{T} \sum_{i=1}^{K} \sum_{j=1}^{M} J(Q_{i,j}^t | \theta)$$

$$= \begin{cases} \frac{1}{2}(Q_{i,j}^t - \max_j Q_{i+1,j}^t)^2, & if \quad i < K, j = a_i^t \\ \frac{\beta}{2}\left[Q_{i,j}^t - (r^t + \gamma \max_j Q_{1,j}^{t+1})\right]^2, \\ & if \quad i = K, j = a_i^t \\ \frac{\lambda}{2}(Q_{i,j}^t - 0)^2, & if \quad j \neq a_i^t \end{cases}$$
$$(5)$$

where $Q_{i,j}^t$ is the Q-value of the $j$-th bin in $Q_i^t$, which is outputted by our Mamba-based Q-Learner $\pi_\theta$, with $[s^t, token(a_{i-1}^t)]$ as input. The first two branches in Eq. (5) are TD errors following the Bellman backup for decomposed Q-function (as described in Eq. (2)). We additionally add a weight $\beta$ (we set $\beta = 10$ in this paper) on the last action dimension to reinforce the learning on this dimension. As described in Eq. (2), the other action dimensions are updated by the inverse maximization operation, so ensuring the accuracy of the Q-value in the last action dimension helps secure the accuracy of the other dimensions. The last branch in Eq. (5) is the conservative regularization introduced in representative offline RL method CQL (Kumar et al., 2020), which is used to relieve the over-estimation due to the distribution shift. Here, the Q-values of action bins that are not selected in the trajectory $\tau$ ($j \neq a_i^t$) are

regularized to 0, which is the lower bound of the Q-values in optimization. This would accelerate the learning of the TD error. We set $\lambda = 1$ in this paper to strike a good balance.

### 4.4. Mamba-based Q-Learner

Existing MetaBBO works primarily struggle in learning meta-level policy with massive joint-action space, which is the configuration space $A : \{A_1, A_2, ..., A_K\}$ associated by $K$ hyper-parameters of the low-level algorithm $A$. To relieve this learning difficulty, we introduce Q-function decomposition strategy as described in Section 3.1. For each hyper-parameter $A_i$ in $A$, we represent its Q-function as a discretized value function $Q_i = \{Q_{i,1}, Q_{i,2}, ..., Q_{i,M}\}$, where $M$ is a pre-defined number of action bins for all $A_i$ in $A$ ($M = 16$ in this paper). For any $A_i$ which takes values from a continuous range, we uniformly discretize the value range into $M$ bins to make universal representation across all $A_i$. By doing this, we turn the MDP in MetaBBO into a sequence prediction problem: we regard predicting each $Q_i$ as a single decision step, then at time step $t$ of the low-level optimization, the complex associated configuration $a_{1:K}^t$ of $A$ can be sequentially decided. We further design a Mamba-based Q-Learner model to assist sequence modeling of decomposed Q-functions. The overall workflow of the Mamba-based Q-Learner is illustrated in Figure 1. We next elaborate elements in the figure with their motivations.

**Optimization state** $s^t$. In MetaBBO, optimization state $s^t$ profiles two types of information: the properties of the optimization problem to be solved and the low-level optimization progress. In Q-Mamba, we construct the optimization state $s^t$ similar with latest MetaBBO methods (Ma et al., 2024b; Chen et al., 2024; Li et al., 2024b). Concretely, at each time step $t$ in the low-level optimization, an optimization state $s^t \in \mathbb{R}^9$ is obtained by calling a function $cal\_state()$. The first 6 dimensions are statistical features about the population distribution, objective value distribution, etc., which provide the problem property information. The last 3 dimensions are temporal features describing the low-level optimization progress. We leave the calculation details of $s^t$ in Appendix B.

**Tokenization of action bins.** We represent the $M = 16$ action bins of each hyper-parameters $A_i$ in $A$ by 5-bit binary coding: $00000 \sim 01111$. Besides, since we sequentially predict the Q-function for $A_1$ to $A_K$, we additionally use 11111 as a *start* token to activate the sequence prediction. We have to note that for an algorithm $A$, some of its discrete hyper-parameters might hold less than $M$ action bins. For this case, we only use the first several tokens to represent the action bins in these hyper-parameters. In the rest of this paper, we use $token(a_i^t)$ to denote the binary coding of the action bin selected for $A_i$ at time step $t$ of the low-level optimization. The Mamba-based Q-learner auto-regressively

outputs the Q-function values $Q_i^t$ for each $A_i$ in $A$.

**Mamba block.** To obtain $Q_i^t$, the first step is to prepare the input as the concatenation of the optimization state $s^t$ and the previously selected action bin token $token(a_{i-1}^t)$. Then, we apply a Mamba block with the computation described in Section 3.2. It receives the hidden state $h_{i-1}^t$ and the embedding feature $\mathbb{E}_i^t$ and outputs the decision information $\mathbb{O}_i^t$ and hidden state $h_i^t$. $\mathbb{O}_i^t$ is used to parse Q-function $Q_i^t$ and $h_i^t$ is used for next decision step as follows:

$$\mathbb{O}_i^t, h_i^t = \text{mamba\_block}([s^t, token(a_{i-1}^t)], h_{i-1}^t | W_{mamba}),$$
(6)

where $W_{mamba}$ denotes all learnable parameters in Mamba, which includes the state transition parameters $A$, $B$ and $C$, the parameters of discretization step matrix, and time-varying mapping parameters for the state transition parameters. In this paper we use the mamba-block in Mamba repo[1], with default settings. To obtain $\mathbb{O}_1^t$, the last hidden state of time step $t - 1$, $h_K^{t-1}$ is used. The motivation of using Mamba is that: a) MetaBBO task features long-sequence process that involves thousands of decision steps since there are hundreds of optimization steps and $K$ hyper-parameters to be decided per optimization step. Mamba is adopted since it parameterizes the dynamic parameters as functions of input state token, which facilitate flexible learning of long-term and short-term dependencies from historical state sequence (Ota, 2024). b) Mamba equips itself with hardware-aware I/O computation and a fast parallel scan algorithm: PrefixSum (Blelloch, 1990), which allows Mamba has the same memory efficiency as highly optimized FlashAttention (Dao et al., 2022).

**Q-value head.** The Q-value head parses the decision information $\mathbb{O}_i^t$ into the decomposed Q-function $Q_i^t$ through a linear mapping layer.

$$Q_i^t = \sigma(\text{Linear}(\mathbb{O}_i^t | W_{head}, b_{head})))$$
(7)

Here, $\sigma$ is Leaky ReLU activation function, $W_{head} \in \mathbb{R}^{16 \times 16}$ and $b_{head} \in \mathbb{R}^{16}$ are weights and bias. When we obtain $Q_i^t$, we select the action bin with the maximum value for hyper-parameter $A_i$: $a_i^t = \arg\max_j Q_{i,j}^t$, and use $token(a_i^t)$ for inferring the decomposed Q-function $Q_{i+1}^t$ of next decision step. Once the action bins of all hyper-parameters $A_1 \sim A_K$ have been decided, the algorithm $A$ parse all selected action bins to concrete hyper-parameter values and then use them to optimize the problem for one step and obtains the optimization state $s^{t+1}$ from the updated solution population (detailed in Appendix C). To summarize, in Q-Mamba, the meta-level policy $\pi_\theta$ is the Mamba-based Q-Learner, of which the learnable parameters $\theta$ includes $\{W_{mamba}, W_{head}, b_{head}\}$. To meta-train the Q-Mamba, we use AdamW with a learning rate $5e - 3$ to minimize the ex-

---
[1] https://github.com/state-spaces/mamba

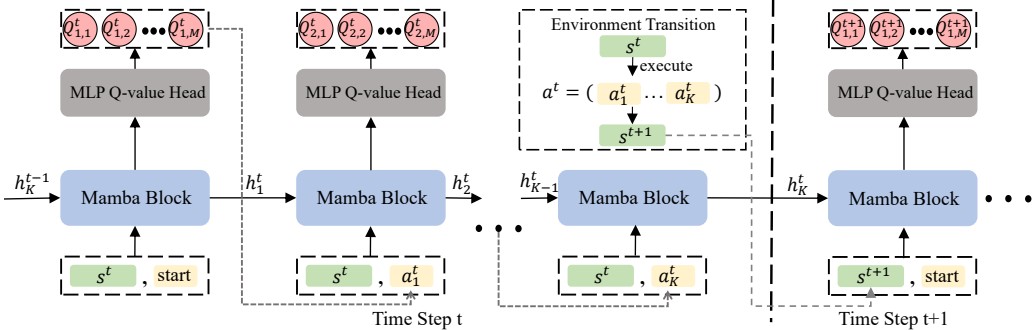

*Figure 1.* The workflow of the Mamba-based Q-Learner. The forward process of the neural network is similar with the Recurrent Neural Network. At each time step, the Q-function of each decomposed action dimension is outputted by conditioning the current state and selected action bins of the previous action dimensions. The environment transition is executed once all action dimensions are outputted.

pectation training objective $\mathbb{E}_{\tau \in \mathbb{C}} J(\tau|\theta)$. After the training, the learned $\pi_\theta$ can be directly used to control $A$ for unseen problems: either those within the same problem distribution $P$ or totally out-of-distribution ones.

# 5. Experimental Results

In the experiments, we aim to answer the following questions: a) How Q-Mamba performs compared with the other online/offline baselines? b) Can Q-Mamba be zero-shot to more challenging realistic optimization scenario? c) How important are the key designs in Q-Mamba?

## 5.1. Experiment Setup

**Training dataset.** We first sample 3 low-level BBO algorithms from the algorithm space constructed in ConfigX (Guo et al., 2024b), which are three evolutionary algorithms including 3, 10 and 16 hyper-parameters, showing different difficulty-levels for MetaBBO methods. We introduce their algorithm structures in Appendix D.1. The problem distribution selected for the training is the *CoCo BBOB Testsuites* (Hansen et al., 2021), which contains 24 basic synthetic functions with diverse properties. We denote it as $P_{bbob}$. We divide it into 16 problem instances for training and 8 problem instances for testing. These functions range from $5 \sim 50$-dimensional, with random shift and rotation on decision variables. More details are provided in Appendix D.2. Based on the 16 training functions, we create a E&E Datasets for each BBO algorithm following the procedure described in Section 4.2. For online MetaBBO baselines, we train them on each low-level algorithm to optimize the training functions. For offline baselines including our Q-Mamba, we train them on each E&E Dataset. Note that the total optimization steps for the low-level optimization is set as $T = 500$.

**Baselines.** We compare a wide range of baselines to obtain comprehensive and significant experimental observations. Concretely, we compare three **online MetaBBO baselines**:

RLPSO (Wu & Wang, 2022) that uses simple MLP architecture for controlling low-level algorithms. LDE (Sun et al., 2021) that facilitates LSTM architecture for sequential controlling low-level algorithms using temporal optimization information. GLEET (Ma et al., 2024b) that uses Transformer architecture for mining the exploration-exploitation tradeoff during the low-level optimization. These three baselines are all trained to output associated configuration without decomposition as our Q-Mamba. Since there is no offline MetaBBO baseline yet, we examine the learning effectiveness of Q-Mamba by comparing it with a series of **offline RL baselines**: DT (Chen et al., 2021), DeMa (Dai et al., 2024a), QDT (Yamagata et al., 2023) and QT (Hu et al., 2024) are four baselines that apply conditional imitation learning on the E&E dataset, where the state, actions and reward in E&E dataset are transformed into RTG tokens, state tokens associated action tokens for supervised sequence-to-sequence learning. The differences are: DT and DeMa follow naive paradigm with Transformer and Mamba architecture respectively. QDT and QT train a separate Q-value predictor during the sequence-to-sequence learning, which relabels the RTG signal to attain policy improvement. Q-Transformer (Chebotar et al., 2023) shows similar Q-value decomposition scheme as our Q-Mamba, while the neural network architecture is Transformer. The settings of all baselines follow their original papers, except that the training data is the prepared three E&E datasets. To ensure the fairness of the comparison, all baselines are trained for 300 epochs with batch size 64.

**Performance metric.** We adopt the accumulated performance improvement $Perf(A, f|\pi_\theta)$ for measuring the optimization performance of the compared baselines and our Q-Mamba. Given a MetaBBO baseline $\pi_\theta$, the corresponding low-level algorithm $A$ and an optimization problem instance $f$, the accumulated performance improvement is calculated as the sum of reward feedback at each optimization step $t$: $Perf(A, f|\pi_\theta) = \sum_{t=1}^{T} r^t$. The reward feedback is calculated as the relative performance improvement between two consecutive optimization steps: $r^t = \frac{f^{*,t-1} - f^{*,t}}{f^{*,0} - f^*}$, where

*Table 1.* Performance comparison between Q-Mamba and other online/offline baselines. All baselines are tested on unseen problem instances within the training distribution $P_{bbob}$. We additionally present the averaged training/inferring time of all baselines in the last row.

| | Online | | | Offline | | | | | |
|---|---|---|---|---|---|---|---|---|---|
| | RLPSO (MLP) | LDE (LSTM) | GLEET (Transformer) | DT | DeMa | QDT | QT | Q-Transformer | Q-Mamba |
| $Alg0$ $K=3$ | 9.855E-01 ±9.038E-03 | 9.563E-01 ±1.830E-02 | 9.616E-01 ±3.110E-03 | 9.325E-01 ±2.680E-02 | 9.492E-01 ±2.467E-02 | 9.683E-01 ±2.131E-02 | 9.729E-01 ±1.934E-02 | 9.666E-01 ±1.975E-02 | **9.889E-01** ±**7.779E-03** |
| $Alg1$ $K=10$ | 9.953E-01 ±3.322E-03 | 9.877E-01 ±1.118E-02 | 9.938E-01 ±2.834E-03 | 6.764E-01 ±1.193E-01 | 9.015E-01 ±1.688E-02 | 9.917E-01 ±5.454E-03 | 9.955E-01 ±3.115E-03 | 9.951E-01 ±3.487E-03 | **9.973E-01** ±**2.441E-03** |
| $Alg2$ $K=16$ | 9.914E-01 ±4.497E-03 | 9.904E-01 ±6.306E-03 | 9.910E-01 ±5.846E-03 | 8.706E-01 ±3.951E-02 | 9.159E-01 ±2.015E-02 | 9.919E-01 ±7.456E-03 | 9.926E-01 ±6.874E-03 | 9.895E-01 ±6.754E-03 | **9.950E-01** ±**9.981E-03** |
| Avg Time | 28h / 11s | 28h / 12s | 25h / 13s | 13h / 10s | 12h / 10s | 20h / 12s | 20h / 12s | 16h / 11s | 13h / 10s |

$f^{*,t}$ is the objective value of the best found solution until time step $t$, $f^*$ is the optimum of $f$. The maximal accumulated performance improvement is 1 when the optimum of $f$ is found. Note that $f^*$ is unknown for training problem instances, we instead use a surrogate optimum for it, which can be easily obtained by running an advanced BBO algorithm on the training problems for multiple runs.

## 5.2. In-distribution Generalization

After training, we compare the generalization performance of our Q-Mamba and other baselines on the 8 problem instances in $P_{bbob}$ which have not been used for training. Specifically, for each baseline and each low-level algorithm, we report in Table 1 the average value and error bar of the accumulated performance improvement $Perf(\cdot)$ across the 8 tested problems and 19 independent runs. We additionally present the average training time and inferring time (time consumed to complete a DAC process for BBO algorithm $A$ on a given optimization problem) for each baseline in the last row. The results show following key observations:

i) **Q-Mamba v.s. Online MetaBBO.** Surprisingly, Q-Mamba achieves comparable/superior optimization performance to online baselines RLPSO, LDE and GLEET, while showing clear advantage in training efficiency. The performance superiority might origins from the Q-function decomposition scheme in Q-Mamba, which avoids searching DAC policy from the massive associated configuration space as these online baselines. The improved training efficiency validates the core motivation of this work. By learning from the offline E&E dataset, Q-Mamba reduces the training budget to less than half of those online baselines. his is especially appealing for BBO scenarios where the simulation is expensive or time-consuming.

ii) **Q-Mamba v.s. DT&DeMa.** We observe that DT and DeMa hold similar training efficiency with our Q-Mamba. The difference between them and Q-Mamba is that they generally imitates the trajectories in E&E dataset by predicting the tokens autoregressively. Results in the table show the performances of DT and DeMa are quite unstable (with large variance). In opposite, our Q-Mamba allows policy improvement during the sequence learning, which shows better learning convergence and effectiveness than the conditional

imitation-learning based offline RL. We note that this observation is limited within MetaBBO domain in this paper, further validation tests are expected to examine Q-Mamba on other RL tasks, which we leave as future works.

iii) **Q-Mamba v.s. QDT&QT.** Comparing QDT&QT with DT, a tradeoff between the learning effectiveness and training efficiency. QDT&QT both propose additional Q-value predictor, which is subsequently used to relabel the RTG tokens in offline dataset. Although relabeling RTG with learned Q-value allows for policy improvement during the conditional imitation learning, additional training resource is introduced (20h v.s. 13h). Nevertheless, QDT&QT, as the online MetaBBO baselines, searches DAC policy from the massive associated action space. Compared with them, our Q-Mamba achieves not only superior optimization performance since we learn easier decomposed Q-function for each hyper-parameter in BBO algorithm, but also similar training efficiency with DT since the hardware-friendly computation and parallel scan algorithm in Mamba.

iv) **Q-Mamba v.s. Q-Transformer.** While our Q-Mamba shares the Q-function decomposition scheme with Q-Transformer, a major novelty we introduced is the Mamba architecture and the corresponding weighted Q-loss function. The superior performance of Q-Mamba to the Q-Transformer possibly roots from the linear time invariance (LTI) of Transformer, which presents fundamental limitation in selectively utilizing short-term or long-term temporal dependencies in the long Q-function sequence. In contrast, Mamba architecture holds certain advantages: it allows Q-Mamba selectively remembers or forgets historical states based on current token. Mamba architecture achieves this through parameterizing the dynamic parameters in Eq. (3) as functions of input state tokens.

## 5.3. Out-of-distribution Generalization

We have to note that the core motivation of MetaBBO is generalizing the meta-level policy trained on simple and economic BBO problems towards complex realistic BBO scenarios. We hence examine the generalization performance of Q-Mamba and three online MetaBBO baselines on a challenging scenario: neuroevolution (Such et al., 2017) tasks. In a neuroevolution task, a BBO algorithm is used to evolve

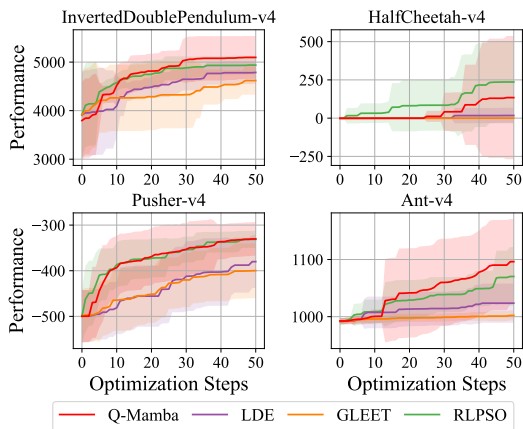

*Figure 2.* Zero shot performance of Q-Mamba and online MetaBBO baselines on neuroevolution tasks.

a population of neural networks according to their performance on a specific machine learning task, i.e., classification, robotic control (Galván & Mooney, 2021). Specifically, we consider continuous control tasks in Mujoco (Todorov et al., 2012). We optimize a 2-layer MLP policy for each task by Q-Mamba and other baselines trained for controlling $Alg0$ on $P_{bbob}$. To align with the challenging condition in realistic BBO tasks, we only allow the low-level optimization involves a small population (10 solutions) and $T = 50$ optimization steps. We present the average optimization curves across 10 independent runs in Figure 2. The results underscore the potential positive impact of Q-Mamba for MetaBBO domain: a) while only trained on synthetic problems with at most 50 dimensions, our Q-Mamba is capable of optimizing the MLP polices which hold thousands of parameters in neuroevolution tasks. b) compared to online MetaBBO baselines, Q-Mamba is capable of learning effective policy with comparable generalization performance, while only consuming less than half training resources.

### 5.4. Ablation Study

**Coefficients in Q-loss.** In Q-Mamba, a key design that ensures the learning effectiveness is the proposed compositional Q-loss in Eq. (5), which calculates a bellman backup on the decomposed Q-function sequence first and applies conservative regularization on out-of-distribution action bins. As shown in Table 2, when $\lambda = 0$, the training objective in Eq. (5) turns into the Bellman backup without conservative regularization. The performance degradation under this setting reveals the importance of the conservative term for relieving the distribution shift caused by offline leaning. When $\beta = 1$, the training objective would not focus on the Q-value prediction of the last action dimension, which in turn interferes the prediction of other action dimensions through the inverse maximization operation in Eq. (2). A setting with $\lambda = 1$ and $\beta = 10$ generally ensures the overall learning effectiveness.

*Table 2.* Importance analysis on $\lambda$ and $\beta$ in compositional Q-loss function.

|  | $\lambda = 0$ | $\lambda = 1$ | $\lambda = 10$ |
|---|---|---|---|
| $\beta = 1$ | 9.756E-01 ±1.570E-02 | 9.828E-01 ±1.203E-02 | 9.855E-01 ±1.192E-02 |
| $\beta = 10$ | 9.833E-01 ±1.424E-02 | **9.889E-01** **±7.780E-03** | 9.857E-01 ±1.134E-02 |

*Table 3.* Performance of Q-Mamba under different proportion of exploitation data with good quality.

| $\mu$ | 0 | 0.25 | 0.5 | 0.75 | 1 |
|---|---|---|---|---|---|
| Perf. | 9.832E-01 ±1.264E-02 | 9.874E-01 ±6.489E-03 | **9.889E-01** **±7.780E-03** | 9.793E-01 ±1.614E-02 | 9.834E-01 ±9.692E-03 |

**Data ratio in E&E dataset.** Another key design in Q-Mamba is the construction of E&E dataset. When collecting DAC trajectories to construct it, we set a data mixing ratio $\mu$ which controls the proportion of exploitation data and exploration data. When $\mu = 0$, all trajectories come from a random configuration policy, which provides exploratory experiences with relatively low quality. When $\mu = 1$, all trajectories come from the well-performing MetaBBO baselines, which provides at least sub-optimal DAC experiences with high quality. The results in Table 3 reveal that mixing these two types of data equally ($\mu = 0.5$) might enhance Q-Mamba's learning effectiveness by leveraging the rich historical experiences from both exploration and exploitation. This actually follows a common sense that increasing data diversity could reduce the training bias in offline learning.

## 6. Conclusion

In this paper, we propose Q-Mamba as a novel offline learning-based MetaBBO framework which improves both the effectiveness and the training efficiency of existing online leaning-based MetaBBO methods. To achieve this, Q-Mamba decomposes the associated Q-function for the massive configuration space into sequential Q-functions for each configuration. We further propose a Mamba-based Q-Learner for effective sequence learning tailored for such Q-function decomposition mechanism. By incorporating with a large scale offline dataset which includes both the exploration and exploitation trajectories, Q-Mamba consumes less than half training time of existing online baselines, while achieving strong control power across various BBO algorithms and diverse BBO problems. Our framework does have certain limitation. Q-Mamba is trained for a given BBO algorithm and requires re-training for other algorithms. An effective algorithm feature extraction mechanism may enhance Q-Mamba's co-training on various algorithms. We mark this as an important future work. At last, we hope Q-Mamba could serve as a meaningful preliminary study, providing first-hand experiences on integrating efficient offline learning pipeline into MetaBBO systems.

## Acknowledgements

This work is supported in part by Guangdong Provincial Natural Science Foundation for Outstanding Youth Team Project (Grant No. 2024B1515040010), in part by National Natural Science Foundation of China (Grant No. 62276100), in part by Guangzhou Science and Technology Elite Talent Leading Program for Basic and Applied Basic Research (Grant No. SL2024A04J01361). This research is also supported by National Research Foundation, Singapore under its AI Singapore Programme (AISG Award No: AISG3-RP-2022-031).

## Impact Statement

This paper presents work whose goal is to advance the field of Evolutionary Computation and Black-Box-Optimization. In particular, it explores the emerging topic Learning to Optimize where meta-learning for automated algorithm design has been widely discussed and studied. A potential impact of this work highly aligns with the impact of automated algorithm design for human society, that is, reducing the design bias introduced by human experts in existing human-crafted BBO algorithms to unleash their optimization performance over industrial-level application and important scientific discovery process. Besides, since this paper provides a pioneering exploration on effectiveness of offline learning in learning-based automated algorithm design, it potentially impacts existing online learning paradigms, hence potentially accelerating the development in this area.

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

# A. Proof of Q-function Decomposition

To show that transforming MDP into a per-action-dimension form still ensures optimization of the original MDP, we show that optimizing the Q-function for each action dimension is equivalent to optimizing the Q-function for the full action. We omit the time step superscript $t$ for the ease of presentation.

If we consider apply full action $a_{1:K}$ at the current state $s$ to transit to the next step state $s'$. The Bellman update of the optimal Q-function could be written as:

$$
\begin{aligned}
\max_{a_{1:K}} Q(a_{1:k}|s) &= \max_{a_{1:K}} \left[ R(s, a_{1:K}) + \gamma \max_{a_{1:K}} Q(a_{1:K}|s') \right] \\
&= R(s, a_{1:K}^*) + \gamma \max_{a_{1:K}} Q(a_{1:K}|s')
\end{aligned}
\tag{8}
$$

where $R(\cdot)$ is the reward we get after executing the full action $a_{1:K}$. Under the Q-function decompostion, the Bellman update of the optimal Q-function for each action dimension $a_i$ is:

$$
\begin{aligned}
\max_{a_i} Q(a_i|s, a_{1:i-1}^*) &= \max_{a_i} \left[ \max_{a_{i+1}} Q(a_{i+1}|s, a_{1:i}^*) \right] \\
&= \max_{a_i} \left[ \max_{a_{i+1}} \left( \max_{a_{i+2}} Q(a_{i+2}|s, a_{1:i+1}^*) \right) \right] \\
&= \cdots \\
&= R(s, a_{1:K}^*) + \gamma \max_{a_1} Q(a_1|s') \\
&= R(s, a_{1:K}^*) + \gamma \max_{a_1} \left[ \max_{a_2} Q(a_2|s', a_1) \right] \\
&= \cdots \\
&= R(s, a_{1:K}^*) + \gamma \max_{a_{1:K}} Q(a_{1:K}|s')
\end{aligned}
\tag{9}
$$

Here the first two lines are the inverse maximization operation as described in Section 3.1, the fourth line is the Bellman update for the last action dimension. The last three lines also follow the inverse maximization operation. By comparing Eq. (8) and Eq. (9) we prove that optimizing the decomposed Q-function consistently optimizes the original full MDP.

# B. Optimization State Design

The formulation of the optimization state features is described in Table 4. States $s_{\{1 \sim 6\}}$ are optimization problem property features which collectively represent the distributional features and the statistics of the objective values of the current candidate population. Specifically, state $s_1$ represents the average distance between each pair of candidate solutions, indicating the overall dispersion level. State $s_2$ represents the average distance between the best candidate solution in the current population and the remaining solutions, providing insights into the convergence situation. State $s_3$ represents the average distance between the best solution found so far and the remaining solutions, indicating the exploration-exploitation stage. State $s_4$ represents the average difference between the best objective value found in the current population and the remaining solutions, and $s_5$ represents the average difference when compared with the best objective value found so far. State $s_6$ represents the standard deviation of the objective values of the current candidates. Then, states $s_{\{7,8,9\}}$ collectively represent the time-stamp features of the current optimization progress. Among them, state $s_7$ denotes the current process, which can inform the framework about when to adopt appropriate strategies. States $s_8$ and $s_9$ are measures for the stagnation situation.

*Table 4.* Formulations of state features.

|  | States | | Notes |
|---|---|---|---|
| **Problem Property** | $s_1^t$ | $\underset{x_i, x_j \in X^t}{mean} \lVert x_i - x_j \rVert_2$ | Average distance between any pair of individuals in current population. |
|  | $s_2^t$ | $\underset{x_i \in X^t}{mean} \lVert x_i - x^{*,t} \rVert_2$ | Average distance between each individual and the best individual in $t$-th generation. |
|  | $s_3^t$ | $\underset{x_i \in X^t}{mean} \lVert x_i - x^* \rVert_2$ | Average distance between each individual and the best-so-far solution. |
|  | $s_4^t$ | $\underset{x_i \in X^t}{mean}(f(x_i) - f(x^*))$ | Average objective value gap between each individual and the best-so-far solution. |
|  | $s_5^t$ | $\underset{x_i \in X^t}{mean}(f(x_i) - f(x^{*,t}))$ | Average objective value gap between each individual and the best individual in $t$-th generation. |
|  | $s_6^t$ | $\underset{x_i \in X^t}{std}(f(x_i))$ | Standard deviation of the objective values of population in $t$-th generation, a value equals 0 denotes converged. |
| **Optimization Progress** | $s_7^t$ | $(T - t)/T$ | The potion of remaining generations, $T$ denotes maximum generations for one run. |
|  | $s_8^t$ | $st/T$ | $st$ denotes how many generations the algorithm stagnates improving. |
|  | $s_9^t$ | $\begin{cases} 1 & \text{if } f(x^{*,t}) < f(x^*) \\ 0 & \text{otherwise} \end{cases}$ | Whether the algorithm finds better individual than the best-so-far solution. |

## C. Action Discretization and Reconstruction

Given the $M = 16$ bins of Q values $Q_i^t$ for the $i$-th action, if the $i$-th hyper-parameter $A_i$ of the low-level algorithm is in continuous space, we first uniformly discretize the space into $M$ bins: $\hat{A}_i = \{A_{i,1}, A_{i,2}, \cdots, A_{i,M}\}$ where $A_{i,1}$ and $A_{i,M}$ are the lower and upper bounds of the space. Then we use the action $a_i^t$ obtained by $a_i^t = \arg\max_j Q_{i,j}^t$ as an index and assign the value of the $i$-th hyper-parameter $A_i$ with $A_i = \hat{A}_i[a_i^t]$. If the hyper-parameter is in discrete space $\hat{A}$ with $m_i \leq M$ candidate choices, the action $a_i^t$ is obtained by $a_i^t = \arg\max_{j \in [1, m_i]} Q_{i,j}^t$ and the value of the $i$-th hyper-parameter is $\hat{A}[a_i^t]$.

After the value of all hyper-parameters are decided, the algorithm $A$ takes a step of optimization with the hyper-parameters and return the next state from the updated population.

## D. Experiment Setup

### D.1. Backend Algorithm Generalization

In this paper, we randomly sample 3 algorithms with action space dimensions 3, 10 and 16 from the algorithm construction space proposed in ConfigX (Guo et al., 2024b), which contains various operators with controllable parameters such as the mutation and crossover operators from DE (Storn & Price, 1997), PSO update rules (Kennedy & Eberhart, 1995), crossover and mutation operators from GA (Holland, 1992). Operators without controllable parameters such as selection and population reduction operators are also included. Then, to get an algorithm with $n$ controllable actions, we keep randomly sampling algorithms from the algorithm construction space and eliminating the algorithms that are not meeting

---

**Algorithm 1** Pseudo code of $Alg0$

---

1: **Input**: Optimization problem $f$, optimization horizon $T$, Meta-level agent $\pi$.
2: **Output**: Optimal solution $x^* = \underset{x \in X}{\arg\min} f(x)$.
3: Uniformly initialize a population $X_1$ with shape $NP_1 = 100$ and evaluate it with problem $f$;
4: **for** $t = 1$ **to** $T$ **do**
5:      Receive the 3 action values $a_t = \{F1, F2, Cr\}$ from the agent $\pi$;
6:      Generate $X_t'$ by using DE/current-to-rand/1 (Eq. (10)) on $X_t$;
7:      Apply Exponential crossover (Eq. (11)) on $X_t$ and $X_t'$ to get $X_t''$;
8:      Clip the values beyond the search range in $X_t''$;
9:      Calculate $f(X_t'')$;
10:      Compare $f(X_t)$ and $f(X_t'')$, select the better solutions to generate $X_{t+1}$;
11: **end for**

---

the requirement, until the algorithm with $n$ controllable actions is obtained. The uncontrollable hyper-parameters of the algorithm such as the initial population sizes are randomly determined.

$Alg0$ (as shown in Algorithm 1) is DE/current-to-rand/1/exponential (Storn & Price, 1997) with Linear Population Size Reduction (LPSR) (Tanabe & Fukunaga, 2014). The mutation operator DE/current-to-rand/1 is formulated as:

$$x_i' = x_i + F1(x_{r1} - x_i) + F2(x_{r2} - x_{r3}) \tag{10}$$

where $x_{r.}$ are randomly chosen solutions and $F1, F2 \in [0,1]$ are two controllable parameters. The Exponential crossover operator is formulated as:

$$x_i'' = \begin{cases} x_{i,j}', & \text{if } rand_{k:j} < Cr \text{ and } k \le j \le L + k \\ x_{i,j}, & \text{otherwise} \end{cases} \quad , j = 1, \cdots, Dim \tag{11}$$

where $Dim$ is the solution dimension, $L \in \{1, \cdots, Dim\}$ is a random length, $rand \in [0,1]^{Dim}$ is a random vector, $x_i'$ is the trail solution generated by mutation operator and $Cr \in [0,1]$ is a controllable parameter. At the beginning, a population $X$ with size 100 is uniformly sampled and evaluated. In each optimization generation, given the parameters $F1, F2, Cr$ from the meta-level agent, algorithm applies DE/current-to-rand/1 mutation and Exponential crossover operator on the population to generate the trail solution population $X_t''$. An comparison is conducted between population $X_t$ and $X_t''$ where the better solutions are selected for the next generation population $X_{t+1}$. Finally the worst solutions are removed from $X_{t+1}$ in the LPSR process.

The second algorithm $Alg1$ (as shown in Algorithm 2) is a hybrid algorithm comprising two sub-populations optimized by GA and DE respectively. The population is sampled in Halton sampling (Halton, 1960) and then divided into two sub-populations with sizes 50 and 200. The first GA sub-population uses the Multi-Point Crossover (MPX) (Holland, 1992) and Gaussian mutation (Holland, 1992) accompanying with the Roulette selection (Holland, 1992). MPX crossover is formulated as:

$$x_i' = \begin{cases} x_{r1,j}', & \text{if } rand_j < Cr_1 \\ x_{i,j}', & \text{otherwise} \end{cases} \quad , j = 1, \cdots, Dim \tag{12}$$

where $rand_j \in [0,1]$ are random numbers, $Cr_1$ is a controllable parameter and $x_{r1}$ is a random solution. The sample method of $x_{r1}$ is also a controllable action $Xr_{mpx}$ which can be uniform sampling or sampling with fitness based ranking. The Gaussian mutation is written as:

$$x_i'' = \mathcal{N}(x_i', \sigma \cdot (ub - lb)) \tag{13}$$

where $ub$ and $lb$ are the upper and lower bounds of the search space and $\sigma \in [0,1]$ is a controllable parameter. The mutated solution is then bound controlled using a composite bound controlling operator which contains 5 bound controlling methods: "clip", "rand", "periodic", "reflect" and "halving" (Kadavy et al., 2023), the selection of the bounding methods is a controllable parameter $bc_1 \in [0,4]$. Besides, the GA sub-population adopts the LPSR technique from initial population size 50 to the final size 10.

In the second DE sub-population, DE/best/2 (Storn & Price, 1997) mutation and binomial (Storn & Price, 1997) crossover are used. DE/best/2 is formulated as:

$$x_i' = x^* + F1 \cdot (x_{r1} - x_{r2}) + F2 \cdot (x_{r3} - x_{r4}) \tag{14}$$

---

**Algorithm 2** Pseudo code of $Alg1$

---

1: **Input**: Optimization problem $f$, optimization horizon $T$, Meta-level agent $\pi$.
2: **Output**: Optimal solution $x^* = \underset{x \in X}{\arg\min} f(x)$.
3: Initialize 2 sub-populations $\{X_{1,1}\}$ and $\{X_{2,1}\}$ using Halton sampling with sizes 50 and 200;
4: Evaluate the sub-populations with problem $f$;
5: **for** $t = 1$ **to** $T$ **do**
6:     Receive the 10 action values $a_t$ from the agent $\pi$;
7:     Generate $X_{1,t+1}$ using MPX (Eq. (12)), Gaussian mutation (Eq. (13)) and Roulette selection on $X_{1,t}$;
8:     Generate $X_{2,t+1}$ using DE/best/2 mutation (Eq. (14)) and binomial crossover (Eq. (15));
9:     **for** $i = 1$ **to** 2 **do**
10:         Replace the worst solution in $X_{i,t+1}$ by the best solution in $X_{cm_i,t+1}$;
11:     **end for**
12:     Apply LPSR on sub-population $X_{1,t+1}$;
13: **end for**

---

where $x_{r.}$ are randomly selected solutions, $x^*$ is the best solution, $F1, F2 \in [0,1]$ are controllable parameters.

The Binomial crossover uses a similar process as MPX but introduces a randomly selected index $jrand \in \{1, \cdots, Dim\}$ to ensure the difference between the generated solution and the parent solution:

$$x_i'' = \begin{cases} x_{i,j}', & \text{if } rand_j < Cr_4 \text{ or } j = jrand \\ x_{i,j}', & \text{otherwise} \end{cases} \quad, j = 1, \cdots, Dim \tag{15}$$

where $rand_j$ are random numbers and $Cr_2 \in [0,1]$ is the controllable parameter. The DE sub-population also contains the composite bound control method with controllable parameter $bc_2$.

Besides, both of the two sub-populations employ the information sharing methods which will replace the worst solution in current sub-population $X_i$ with the best solution from $X_{cm_i}$, where $i \in \{1,2\}$ in this algorithm. The parameters $cm_1, cm_2 \in \{1,2\}$ are two controllable parameters for the sharing operator in the two sub-population, respectively. If the action decides to share with itself ($cm_i = i$), the sharing is stopped.

In summary, the action space of $Alg1$ is $\{Cr_1, Xr_{mpx}, \sigma, bc_1, cm_1, F1, F2, Cr_2, bc_2, cm_2\}$ with the shape of 10.

For $Alg2$ (as shown in Algorithm 3), the population sampled in Halton sampling (Halton, 1960) is divided into four sub-populations. The first sub-population uses GA operators MPX (Holland, 1992) crossover formulated in Eq. (12) and Polynomial mutation (Dobnikar et al., 1999) accompanying with the Roulette selection (Holland, 1992). The Polynomial mutation is as follow:

$$x_i'' = \begin{cases} x_i' + ((2u)^{\frac{1}{1+\eta_m}} - 1)(x_i' - lb), & \text{if } u \le 0.5; \\ x_i' + (1 - (2 - 2u)^{\frac{1}{1+\eta_m}})(ub - x_i'), & \text{if } u > 0.5. \end{cases} \tag{16}$$

where $\eta_m \in \{1,2,3\}$ is a controllable parameter, $u \in [0,1]$ is a random number, $ub$ and $lb$ are the upper and lower bound of the search range.

The second sub-population uses SBX crossover (Deb et al., 1995), Gaussian mutation (Holland, 1992) and Tournament selection (Goldberg & Deb, 1991):

$$x_i' = 0.5 \cdot [(1 \mp \beta)x_i + (1 \pm \beta)x_{r1}], \text{ where } \beta = \begin{cases} (2u)^{\frac{1}{1+\eta_c}} - 1, & \text{if } u \le 0.5; \\ (\frac{1}{2-2u})^{\frac{1}{1+\eta_c}}, & \text{if } u > 0.5. \end{cases} \tag{17}$$

where $\eta_c \in \{1,2,3\}$ is controllable parameter and $u \in [0,1]$ is random number. Similar to MPX, SBX also uses an action $Xr_{sbx}$ to select parent solutions $x_{r1}$. The Gaussian mutation operator formulated in Eq. (13) has controllable parameter $\sigma \in [0,1]$.

The third sub-population is DE/rand/2/exponential (Storn & Price, 1997) where the DE/rand/2 mutation operator is:

$$x_i' = x_{r1} + F1_3(x_{r2} - x_{r3}) + F2_3(x_{r4} - x_{r5}) \tag{18}$$

where $x_{r.}$ are randomly selected solutions and $F1_3, F2_3 \in [0,1]$ are controllable parameters for the third sub-population. The Exponential crossover formulated as Eq. (11) is used in this sub-population with parameter $Cr_3 \in [0,1]$.

---

**Algorithm 3** Pseudo code of $Alg2$

---

1: **Input**: Optimization problem $f$, optimization horizon $T$, Meta-level agent $\pi$.
2: **Output**: Optimal solution $x^* = \underset{x \in X}{\arg\min} f(x)$.
3: Initialize 4 sub-populations $\{X_{i,1}\}_{i=1,2,3,4}$ using Halton sampling with sizes $\{200, 100, 100, 100\}$.
4: Evaluate the sub-populations with problem $f$;
5: **for** $t = 1$ **to** $T$ **do**
6:     Receive the 16 action values $a_t$ from the agent $\pi$;
7:     Generate $X_{1,t+1}$ using MPX (Eq. (12)), Polynomial mutation (Eq. (16)) and Roulette selection on $X_{1,t}$;
8:     Generate $X_{2,t+1}$ using SBX (Eq. (17)), Gaussian mutation (Eq. (13)) and Tournament selection on $X_{2,t}$;
9:     Generate $X_{3,t+1}$ using DE/rand/2 mutation (Eq. (18)), Exponential crossover (Eq. (11)) on $X_{3,t}$;
10:    Generate $X_{4,t+1}$ using DE/current-to-best/1 mutation (Eq. (19)), Binomial crossover (Eq. (15)) on $X_{4,t}$;
11:    **for** $i = 1$ **to** $4$ **do**
12:       Replace the worst solution in $X_{i,t+1}$ by the best solution in $X_{cm_i,t+1}$
13:    **end for**
14: **end for**

---

The last sub-population is DE/current-to-best/1/binomial (Storn & Price, 1997). The mutation operator with parameter $F1_4, F2_4 \in [0, 1]$ is formulated as:

$$x'_i = x_i + F1_4(x^* - x_i) + F2_4(x_{r1} - x_{r2}) \tag{19}$$

where $x^*$ is the best performing solution in the sub-population. The Binomial crossover formulated in Eq. (15) contains a controllable parameter $Cr_4 \in [0, 1]$.

Besides, $Alg2$ conducts the controllable information sharing among the sub-populations where the worst solution in current sub-population $X_{i,g}$ is replaced by the best solution from the target sub-population $X_{cm_i,g}, cm_{\{1,2,3,4\}} \in \{1, 2, 3, 4\}$ are four actions indicating the target sub-population.

Given the 16 actions $\{Cr_1, Xr_{mpx}, \eta_m, \eta_c, Xr_{sbx}, \sigma, F1_3, F2_3, Cr_3, F1_4, F2_4, Cr_4, cm_1, cm_2,$ $cm_3, cm_4\}$, $Alg2$ uses these parameters to configure the mutation and crossover operators and applies them on the 4 sub-populations. Then the information sharing is activated for better exploration. Finally, the next generation population is obtained through the population reduction processes.

## D.2. Train-test split of BBOB Problems

As shown in Table 5, the BBOB testsuite (Hansen et al., 2021) contains 24 different optimization problems with diverse characteristics such as unimodal or multi-modal, separable or non-separable, high conditioning or low conditioning. To maximize the problem diversity of the training problem set and hence empower the agent better generalization ability, we choose the most diverse 16 problem instance for training, whose fitness landscapes in 2D scenario are shown in Figure 3. The rest 8 instances are used as testing set whose 2D landscapes are shown in Figure 4. The dimensions of each problem instances in both training and testing set are randomly chosen from $\{5, 10, 20, 50\}$.

*Table 5.* Overview of the BBOB testsuites.

| | Problem | Functions | Dimensions |
|---|---|---|---|
| Separable functions | $f_1$ | Sphere Function | 50 |
| | $f_2$ | Ellipsoidal Function | 5 |
| | $f_3$ | Rastrigin Function | 5 |
| | $f_4$ | Buche-Rastrigin Function | 10 |
| | $f_5$ | Linear Slope | 50 |
| Functions with low or moderate conditioning | $f_6$ | Attractive Sector Function | 5 |
| | $f_7$ | Step Ellipsoidal Function | 20 |
| | $f_8$ | Rosenbrock Function, original | 10 |
| | $f_9$ | Rosenbrock Function, rotated | 10 |
| Functions with high conditioning and unimodal | $f_{10}$ | Ellipsoidal Function | 10 |
| | $f_{11}$ | Discus Function | 5 |
| | $f_{12}$ | Bent Cigar Function | 50 |
| | $f_{13}$ | Sharp Ridge Function | 10 |
| | $f_{14}$ | Different Powers Function | 20 |
| Multi-modal functions with adequate global structure | $f_{15}$ | Rastrigin Function (non-separable counterpart of F3) | 5 |
| | $f_{16}$ | Weierstrass Function | 20 |
| | $f_{17}$ | Schaffers F7 Function | 50 |
| | $f_{18}$ | Schaffers F7 Function, moderately ill-conditioned | 50 |
| | $f_{19}$ | Composite Griewank-Rosenbrock Function F8F2 | 10 |
| Multi-modal functions with weak global structure | $f_{20}$ | Schwefel Function | 20 |
| | $f_{21}$ | Gallagher's Gaussian 101-me Peaks Function | 20 |
| | $f_{22}$ | Gallagher's Gaussian 21-hi Peaks Function | 10 |
| | $f_{23}$ | Katsuura Function | 20 |
| | $f_{24}$ | Lunacek bi-Rastrigin Function | 20 |
| Default search range: $[-5, 5]^{Dim}$ | | | |

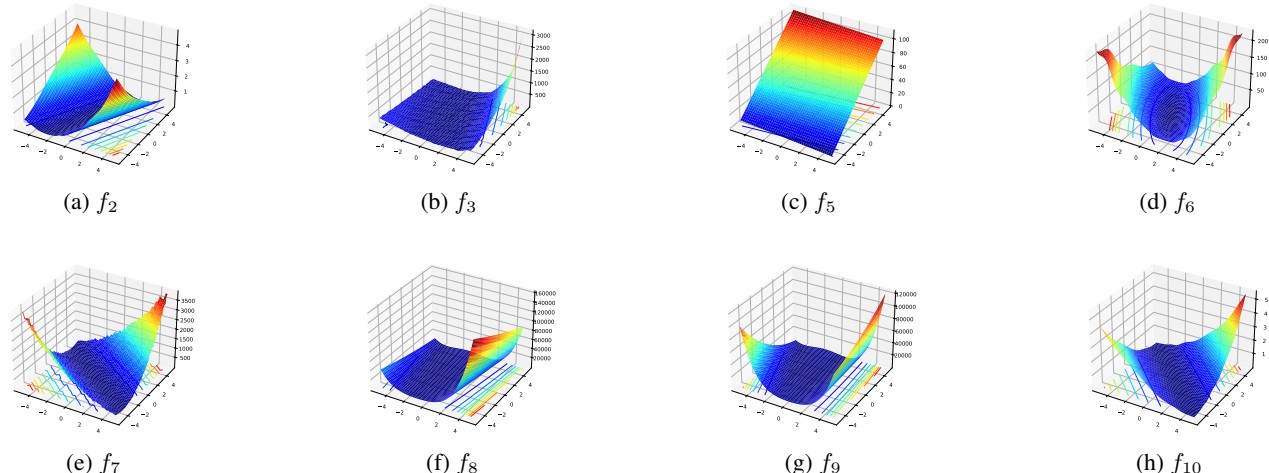

(a) $f_1$    (b) $f_4$    (c) $f_{11}$    (d) $f_{12}$

(e) $f_{13}$    (f) $f_{14}$    (g) $f_{15}$    (h) $f_{16}$

(i) $f_{17}$    (j) $f_{18}$    (k) $f_{19}$    (l) $f_{20}$

(m) $f_{21}$    (n) $f_{22}$    (o) $f_{23}$    (p) $f_{24}$

*Figure 3.* Fitness landscapes of functions in BBOB **train** set when dimension is set to 2.

(a) $f_2$    (b) $f_3$    (c) $f_5$    (d) $f_6$

(e) $f_7$    (f) $f_8$    (g) $f_9$    (h) $f_{10}$

*Figure 4.* Fitness landscapes of functions in BBOB **test** set when dimension is set to 2.

# E. Additional Experimental Results

### E.1. Impact of Action Bin Numbers

As we described in Section 4.4, when the hyper-parameter to be controlled is continuous, the Q-function decomposition scheme have to discretize the continuous space into discrete action bins. The number of action bins determines the control grain. If we use a large number of action bins, the parameter controlling is fine-grained while the action space is increased. If the action bin number is small, the control grain is coarse but the network scale is smaller. In this section we investigate the impact of the action bin numbers on the performance. Concretely, we implement 6 Q-Mamba agents with 16, 32, 64, 128, 256 and 512 bins. Their binary coding of the actions are represented in 5∼10-bits, and the output dimensions of the Q-value head in these baselines are set to 16∼512 accordingly. We train these agents for controlling $Alg0$ on the 16 training BBOB problem instances and then test them on the 8 instance BBOB testing set. The boxplots of the performance values of these baselines over 19 independent runs are presented in Figure 5. The results show that Q-Mamba is compatible with large action bins and fine-grained controlling. However, increasing action bin numbers may not always lead to better performance due to two main reasons: a) the increased network scales and training difficulty. b) for BBO algorithm such as evolutionary algorithms in this paper, their hyper-parameters often show low sensitivity on slight value changes. In this case, increasing number of action bins makes little effect on the target BBO algorithm. In conclusion, the setting of 16 action bins is an ideal choice balancing the control grain and training efficiency, which is used in all trainings of Q-Mamba in our experiments.

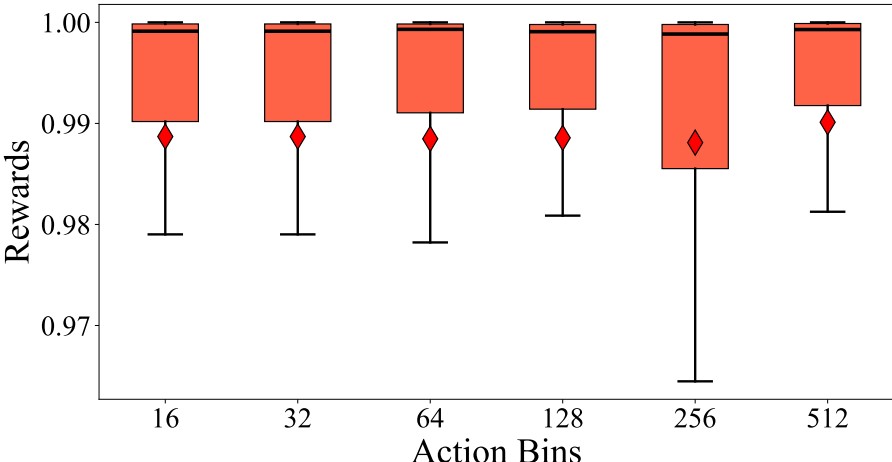

*Figure 5.* The performances of Q-Mamba trained with different action bin granularities.

