# OpenReview forum: "Meta-Black-Box-Optimization through Offline Q-function Learning"
_ICML.cc/2025/Conference — ICML 2025 poster_

### Official Review · Reviewer_TQqs · 2025-02-28

**Overall Recommendation:** 4

**Summary:**

This paper introduces Q-Mamba, a meta-black-box optimization framework integrates offline reinforcement learning and Mamba architecture to achieve effectiveness and efficiency. Q-Mamba is trained on 16 black-box optimization tasks to meta-learn an optimal algorithm configuration, demonstrating comparable or superior performance on black-box optimization tasks and neuroevolution tasks.

## update after rebuttal
All my comments have been addressed. I am also leaning towards acceptance.

**Claims And Evidence:**

The authors demonstrate the efficiency of Q-Mamba by comparing its training/inferring time cost with other baselines. But this demonstration doesn’t rigorously show the efficiency comes from the Mamba architecture, even when compared to the structurally similar Q-Transformer.

**Essential References Not Discussed:**

There are 2 online MetaBBO methods, both of which are trained/tested on CoCo BBOB Testsuites:
[1] Lange, R. T., Schaul, T., Chen, Y., Zahavy, T., Dalibard, V., Lu, C., Singh, S., and Flennerhag, S. Discovering evolution strategies via meta-black-box optimization. In International Conference on Learning Representations, 2023b.
[2] Li, X., Wu, K., Li, Y. B., Zhang, X., Wang, H., and Liu, J. Pretrained optimization model for zero-shot black box optimization. In The Thirty-eighth Annual Conference on Neural Information Processing Systems, 2024b.

**Experimental Designs Or Analyses:**

1)	Experiment Setup
The authors split the CoCo BBOB Testsuites into 16 training functions and 8 testing functions. According to Appendix D.2, the training functions are mostly multi-modal with high conditioning, while the testing functions are mostly unimodal with low conditioning, under such split, the performance of Q-Mamba may not be well evaluated.
2)	Ablation Study
The performance differences between low-level BBO algorithms pre-trained with Q-Mamba and the same algorithms without such pre-training to show the effectiveness of Q-Mamba.

**Methods And Evaluation Criteria:**

The overall method of Q-Mamba framework is adopting offline reinforcement learning and Mamba architecture to ensure training efficiency, and Q-function decomposition scheme for learning effectiveness, which dose make sense.

**Other Comments Or Suggestions:**

It might be worth exploring other ways of train-test splitting on CoCo BBOB Testsuites.

**Other Strengths And Weaknesses:**

Weakness: Q-Mamba appears to lack the capability for hyperparameter self-adaptation, it cannot adaptively adjust the hyperparameters of the low-level BBO algorithm based on the optimization problem at hand.

**Questions For Authors:**

Q1: Did the authors test Q-Mamba on black-box functions as complex as  f16, f21, f22 in CoCo BBOB Testsuites? If so, what were the results?
Q2: According to paragraph Q-value head, subsection 4.4, in training phase, the low-level BBO algorithm optimizes for only one generation on the problem per DAC process. Is this sufficient to evaluate the current hyperparameter configuration?

**Relation To Broader Scientific Literature:**

1)	Meta-Black-Box Optimization: The proposed Q-Mamba framework is an offline MetaBBO method.
2)	Offline Reinforcement Learning: Q-Mamba employs an offline reinforcement learning method at the meta-level.
3)	Over-Estimation Relieving: The conservative Q-learning loss is applied in Q-Mamba to mitigate the overestimation issue caused by distribution shift.
4)	Q-Function Decomposition: Q-Mamba adopts a Q-function decomposition strategy to enhance its learning effectiveness.
5)	Mamba Architecture: Q-Mamba employs the Mamba architecture for long sequence learning.

**Theoretical Claims:**

The only proof in this submission is the proof of Q-function decomposition in Appendix A, which proved the consistency between optimizing the Q-function for each action dimension and optimizing the Q-function for the full action. I didn’t see any issues.

---

> ### Author Rebuttal · Authors · 2025-03-31
>
> We appreciate the reviewer for your valuable comments. We provide reponses as below to address your concerns.
>
> **[Advantages of Q-Mamba]** We would like to first clarify that the core motivation of Q-Mamba is providing an offline learning paradigm for MetaBBO domain, with at least comparable performance with online MetaBBO approaches and significantly reduced training cost. From Table 1 (in-distribution test) and Figure 2 (OOD test), we have validated such aspects. We would like to also argue that by using Mamba architecture to facilitate efficient parallel scan during the training, Q-Mamba actually improve the efficiency of Q-transformer by approximate 20% (13h vs 16h).
>
> **[Train-test split]** We would like to first explain that the train-test split in experiments follows up-to-date MetaBBO methods, where the motivation of such split is to make the meta-level policy learn comprehensively across as many complex problem landscape as possible, hence ensuring a good generalization performance. Meanwhile, we agree with the reviewer that a more uniform train-test split can further demonstrate the learning effectiveness of Q-Mamba. Following this suggestion, we have tried a uniform train-test split (move f16, f17 and f21 from train set to test set, move f6, f8 and f9 from test set to train set) to compare Q-Mamba and other baselines. Due to the space limitation, we provide the results in https://anonymous.4open.science/r/QMamba_review-C0CF/train_test_split.md (due to the narrow rebuttal window, we only provide results on $Alg0$). The results there consistently validate the superior perforamence of Q-Mamba.
>
> **[Ablation on BBO algorithm without Q-Mamba pre-train]** We agree with the reviewer that the learning effectiveness could be further validated by such ablation. Following the suggestion, we have used the same 19 random seeds and $Alg0 \sim Alg2$ to optimize the 8 tested problems. We provide the comparison results in https://anonymous.4open.science/r/QMamba_review-C0CF/Without_pre-train_ablation.md. The results above clearly demonstrate the learning effectiveness of Q-Mamba, which could boost the optimization performance of the backend BBO optimizer.
>
> **[Essential References]** For reference [1][2] the reviewer suggests, we would like to explain that the reason why we only list them as related works rather than compare them as basleines is that Q-Mamba aims at DAC tasks in BBO, while [1][2] explore representing BBO optimizer by NN. We hence chose RLPSO, GLEET and LDE, which are tailored for DAC tasks as baselines to construct the offline data and compare.
>
> **[Self-adaption of Q-Mamba]** We would like to clarify that Q-Mamba inherits the adaption ability of MetaBBO methods. Specifically, as we illustrated in page 5, Figure 1, in each decision step of Q-Mamba, we let the model knows the current optimization status $s^t$, which informs it the optimziation problem and landascape information so that Q-Mamba could adaptovely adjust the hyper-parameters. We hope this could address your concern.
>
> **[Results on f16, f21 and f22]** Since the f16, f21 and f22 are now located at the training dataset in the paper, we have not tested them in in-distribution test in Section 5.2, Table 1. However, we could provide the testing results of the trained Q-Mamba on these training functions, which you can access at https://anonymous.4open.science/r/QMamba_review-C0CF/Performance_f16_f21_f22.md. The results there show that Q-Mamba could significantly boost $Alg0 \sim Alg2$ on these three problem isntances with complex properties.
>
> **[The low-level BBO algorithm optimizes for only one generation on the problem per DAC process…]** This is a very interesting question and we admit that, currently, once Q-Mamba decides a configuration for the low-level optimizer, this configuration is only used for that generation. Q-Mamba need to decide again for the next generation. However, we would like to clarify that this “one config one step” paradigm align with traditional adaptive EAs where in each generation the hyper-parameters are adjusted. We denote this issue as an interesting future work.

---

> > ### Comment · Reviewer_TQqs · 2025-04-02
> >
> > All my questions were clearly addressed and I am happy to recommend acceptance of the paper.

---

> > > ### Author Response · Authors · 2025-04-04
> > >
> > > We sincerely appreciate for your review efforts and recommendation. It is exanctly your valuable and constructive comments that make our paper better!

---

### Official Review · Reviewer_6EuB · 2025-03-11

**Overall Recommendation:** 4

**Summary:**

This paper proposes a Mamba architecture-based meta-black-box optimization framework, Q-Mamba. By conducting offline reinforcement learning on demonstration dataset with diversified behaviours, Q-Mamba achieves competitive or superior performance and efficiency in dynamically configuring BBO algorithms for black-box optimization and Neuroevolution tasks.

## Update after rebuttal
I have carefully read the authors' rebuttal and the feedback has well addressed my questions and concerns. Thus, I would like to insist on the score of accept.

**Claims And Evidence:**

This paper claims that decomposing Q-functions and introducing Mamba architecture would be more efficiency and effective. The comparison results on optimization performance and training/inferring time with online and offline baselines validate the claims on the framework. However, the effectiveness of introducing Mamba architecture and offline RL is not rigorously demonstrated.

**Essential References Not Discussed:**

The discussion on related methods is sound, I didn’t see any essential references not discussed.

**Experimental Designs Or Analyses:**

In Section 5.1. Experiment Setup, the authors split the CoCo BBOB Testsuites into 16 training problem instances and 8 testing problem instances.However, as shown in Appendix D.2, the complexity and difficulty of the training and testing functions are unbalanced. The testing set mostly contains unimodal and low conditioning functions, which may not evaluate well.

In Section 5.2 In-distribution Generalization, comparing Q-Mamba with the pre-trained MetaBBO methods constructing the E&E dataset may further validate the effectiveness of Q-Mamba.

In Section 5.3 Out-of-distribution Generalization, the problem dimensions of the Neuroevolution tasks are not claimed.

In Section 5.4 Ablation Study, the authors investigate the coefficient settings in Q-loss and the data ratio in E&E dataset. Besides these two, the impact of the size of the E&E dataset may also be worth exploring.

**Methods And Evaluation Criteria:**

This paper decomposes Q-functions for each action dimensions and introduces Mamba architecture for efficiency training. The offline RL training on diverse demonstration dataset ensures the effectiveness of the proposed framework. The accumulated rewards in the MDP act as the evaluation criteria, which makes sense.

**Other Comments Or Suggestions:**

Comments:

The comparison with the pre-trained MetaBBO methods constructing the E&E dataset and the impact of the E&E dataset size may be worth exploring.

**Other Strengths And Weaknesses:**

Weakness:

Decomposing Q-functions and making decisions for each action dimensions leads to fine-grained algorithm configuration and prospective better performance, however, it also extends the trajectory lengths and may leads to higher training difficulty and time cost.

**Questions For Authors:**

Q1: In Table 1 Q-Mamba demonstrates faster inference than MLP and LSTM based methods, since the architecture of Mamba may be more complex than MLP and LSTM, what is the reason of the faster inference?

Q2: The optimization state design includes the distances between solutions and objective values, however, the search range and objective value scales of different functions may vary, how to deal with the scale variance in the states?

Q3: It seems that Q-Mamba can also be used for online setting, how it performs on online setting?

Q4: Can authors provide an additional ablation study that validate the effectiveness of the offline-learning and Mamba-architecture?

**Relation To Broader Scientific Literature:**

1. Mamba Architecture: This paper adopts the Mamba architecture for decomposed Q-function decisions.

2. Meta-Black-Box Optimization: This paper proposes a novel MetaBBO framework.

3. Offline Reinforcement Learning: This paper integrates an offline RL method for model training.

**Theoretical Claims:**

In Appendix A, the authors show the proof of Q-function decomposition, which shows that optimizing the Q-function for each action dimension is equivalent to optimizing the Q-function for the full action.

---

> ### Author Rebuttal · Authors · 2025-03-31
>
> We appreciate the reviewer for such comprehensive review and insightful comments. We provide following point-to-point responses to address the concerns in “Experimental Designs Or Analyses”, “Other Strengths And Weaknesses”, “Other Comments Or Suggestions” and “Questions For Authors”.
>
> **[Train-test split]** We agree with the reviewer that a more uniform train-test split can further demonstrate the learning effectiveness of Q-Mamba. Following this suggestion, we have tried a uniform train-test split to compare Q-Mamba and other baselines. Due to the space limitation, we provide the results in https://anonymous.4open.science/r/QMamba_review-C0CF/train_test_split.md. The results there consistently validate the superior perforamence of Q-Mamba.
>
> **[Comparing Q-Mamba with the pre-trained MetaBBO method]** We would like to remind the reviewer that the online baselines we compared in Table 1 are exactly the ones we used to construct our E&E dataset.
>
> **[Problem dimension in OOD testing]** We used two-layer MLP structure (#state_dim, #hidden_dim=64, #action_dim) as the optimizees in these neuroevolution tasks. Since the #state_dim and #action_dim vary across the four Gym enviroments we used, the problem dimension for “InvertedDoublePendulum-v4” is 833, for “HalfCheetah-v4” is 1542, for “Pusher-v4” is 1991 and for “Ant-v4” is 2312. We would add these details into the paper if it was accepted.
>
> **[Impact of E&E dataset size]** Following the suggestion, we have randomly extracted 1K, 3K, 5K trajectories from our previously constructed E&E dataset (10K) to train and test Q-Mamba on $Alg0 \sim Alg2$ respectively. We report the same performance metric as Table 1 of these Q-Mamba variants in https://anonymous.4open.science/r/QMamba_review-C0CF/Dataset_size_impact.md. We observe that, generally, if the dataset includes very narrow experiences data, the performance of Q-Mamba might suffers. A large scale pre-training could ensure the overall learning effectiveness.
>
> **[Training difficulty/cost introduced by the Q-decomposition]** We would like to explain for the reviewer that the training difficulty is significantly reduced by the Q-decomposition. As we elaborated in Section 4.4, line 237-253, applying policy learning on the massive associated configuration space of EAs is challenging, while decomposing this space makes simple q-learning possible, which is usually effective than policy gradient method. The training cost of Q-Mamba is further reduced by our proposed offline learning paradigm.
>
> **[Reason of faster inference]** We note that as we have decribed in Section 4.4, right col, line 248-265, we only used a single default mamba block, with input_dim as 14 (#state_dim=9 + #action_token_dim=5) and output_dim as 16 (the chosen discretization granularity in our paper). With this setting, the total learnable parameters are 5124. For online baselines, the MLP in RLPSO holds 6496 learnable parameters and the LSTM in LDE holds 6560 learnable parameters. This indicates Q-Mamba remains the same complexity. The faster inference comes from the hardware-aware design in Mamba, where it arranges the expanded states and its parameters in GPU SRAM rather than GPU HBM, such “on the fly” computation ensure an acceptable inference efficiency although the sequence is much longer.
>
> **[Numerical scale of state feature]** In Q-Mamba, when we compute state feature, we first min-max normalize the population positions by the searching range of the given problem, then min-max normalize the obejctive values within that population. This facilitate the generalization across various problems.
>
> **[Ablation of online setting Mamba architecture]** Following the suggestion, we have conducted two additional ablation studies:
>
> - First, we remove the thrid case in the training objective (Eq. (5), the CQL regularization in offline setting) and train Q-Mamba in online paradigm. we provide the results in https://anonymous.4open.science/r/QMamba_review-C0CF/Online_Mamba_ablation.md, which indicate that major difference between the online and offline learning of Q-Mamba is the training efficiency. This is due to that in offline setting, we can load the entire trajectory into GPU to facilitate efficient parallel scan of Mamba, hence significantly reducing the training cost.
> - Then we remove the mamba_block in Q-Mamba, leaving only the MLP q-value head, train a Q-Mamba variants with the same training setting. We provide the comparison results in https://anonymous.4open.science/r/QMamba_review-C0CF/Mamba_ablation.md. The results show that Mamba architecture contributes to the performance of Q-Mamba significantly. We would like to remind the reviewer that, in Table 1, the comparison of our Q-Mamba and Q-transformer also provide evidence that Mamba architecture is more suitable for long sequence tasks such as MetaBBO.

---

> > ### Comment · Reviewer_6EuB · 2025-04-02
> >
> > I have carefully read the authors' rebuttal and the feedback has well addressed my questions and concerns. Thus, I would like to insist on the score of 4 (accept) . Thanks.

---

> > > ### Author Response · Authors · 2025-04-04
> > >
> > > We appreciate reviewer #6EuB for the insightful and comprehensive review. We also enjoy the in-depth discussion with the reviewer on some aspects of our Q-Mamba. The valuable suggestions above surely contribute to the scientific integrity of our paper, and we will include them as we have promised if the paper could be accepted!

---

### Official Review · Reviewer_xiEG · 2025-03-13

**Overall Recommendation:** 2

**Summary:**

This paper provides an exploration on effectiveness of offline reinforcement learning in Meta-Black-Box-Optimization to address the training efficiency problem of the online learning paradigms in existing works. The authors transform the DAC task into long sequence decision process and apply a Q-function decomposition scheme with a conservative Q loss. They further use a Mamba-based neural network architecture as the RL agent for long sequence learning capability and efficient training. They conduct both in-distribution and out-of-distribution experiments to show the training efficiency and effectiveness on both in-distribution and out-of-distribution problem instances of their method.

**Claims And Evidence:**

1. The performance metric the authors use in the in-distribution experiment can be misleading, for it depicts the improvement ratio of the best objective value found in the last run to the best objective value in the initial population. If the initial population is not set the same, the performance metric cannot depict how good the final found best objective value is.

2. Although the authors claim their contribution of applying conservative Q loss to address the distributional shift issue in offline RL (which is already applied in Q-transformer[1]), they do not provide any evidence of offline reinforcement learning in Meta Black-Box-Optimization suffering from this distributional shift issue.

3. Although the authors conduct the out-of-distribution experiment to show the generalization performance of their method, they do not compare their method to the state-of-the-art offline RL methods. Moreover, being only compared to zero-shot performance of online Meta-BBO baseline cannot provide solid evidence of the claimed generalization capability of the proposed method, since it is possible that all the methods do poorly in generalization.

[1] Chebotar, Yevgen et al. “Q-Transformer: Scalable Offline Reinforcement Learning via Autoregressive Q-Functions.” Conference on Robot Learning (2023).

**Essential References Not Discussed:**

Generally good

**Experimental Designs Or Analyses:**

Yes, I have checked the experiments.

**Methods And Evaluation Criteria:**

Please see the above comments.

**Other Comments Or Suggestions:**

none

**Other Strengths And Weaknesses:**

1. This paper provides an exploration on effectiveness of offline reinforcement learning in Meta-Black-Box-Optimization to address the training efficiency problem, which provides a foundation for future works.

2. The authors make clear and detailed writing.

3. There are thorough analyses on the experiment results, with their method compared to every single baseline.

4. This paper does not show clear novelty in their method designing: They basically replace the transformer architecture in Q-transformer[1] with the RNN-like Mamba architecture (the Q-function decomposition scheme and the conservative Q loss are almost the same as those in Q-transformer[1]), with no significant improvement on performance and inferring time.

**Questions For Authors:**

1. Can you provide more detailed information about the performance metric you use in the in-distribution experiments (eg how you set the initial population) and the reason you use it?

2. Can you provide numerical results of the best objective function found finally in the in-distribution experiments?

3. Can you provide SOTA offline RL method baselines in the out-of-distribution experiments?

**Relation To Broader Scientific Literature:**

This paper provides an exploration on effectiveness of offline reinforcement learning in Meta-Black-Box-Optimization to address the training efficiency problem of the online learning paradigms in existing works. The proposed method is basically derived from Q-transformer[1], with the transformer architecture replaced by the proposed RNN-like Mamba architecture.

**Theoretical Claims:**

No theory

---

> ### Author Rebuttal · Authors · 2025-03-31
>
> We appreciate the reviewer for the thorough and insightful review. We provide following point-to-point responses to address the concerns in your valuable comments.
>
> **[Performance metric]** We would like to first note that such nomalized metric has been widely used in recent works, e.g., SYMBOL (https://openreview.net/forum?id=vLJcd43U7a , ICLR 24), GLHF (https://openreview.net/forum?id=fWQhXdeuSG , NeurIPS 24) and ConfigX (https://arxiv.org/abs/2412.07507 , AAAI 25). In the rollout process, for each of the 8 tested problem instances, we apply each baseline including our Q-Mamba for controlling $Alg0 \sim Alg2$ to optimize the given problem instances, across 19 independent runs. We would like to clarify that testing a baseline by different initial population is a common practice in BBO testing to measure the performance robustness. As we fix these 19 random seeds for different baselines, these baselines are tested under the same 19 initial conditions hence the fairness of our performance metric is ensured. Besides the normalization in the single step reward provides convenience for presenting average performance across different problem instances for our readers, since various problems show various objective scales. We hope the above explanation could address your concern.
>
> **[Numerical results]** Following your suggestion, we have provided the numerical results of in-distribution testing in https://anonymous.4open.science/r/QMamba_review-C0CF/Numerical_results.md. We provide 3 tables ($Alg0 \sim Alg2$) for each of the 8 tested problems respectively, this is because if we do not normalize them as we did in the paper, these results can not be averaged into one table. We respectifully request the reviewr to check these results, which show that the actual performance superiorty of Q-Mamba is even more larger.
>
> **[Evidence of distribution shift]** We thank the reviewer for this valuable suggestion! Indeed, we should provide more direct evidence to validate there is indeed distribution shift in offline MetaBBO. To this end, we have additionally trained Q-Mamba models under no-CQL setting. In specific, we modify the training objective in Eq. (5) of our paper by removing the q-value regularization term on OOD actions (the thrid case). We provide the performance results in https://anonymous.4open.science/r/QMamba_review-C0CF/no-CQL_ablation.md. The results provide a clear evidence that, if we remove the q-value regularization, the distribution shift could significantly downgrade the performance of Q-Mamba.
>
> **[SOTA offline RLs in OOD]** Following the suggestion, we zero-shot two offline RL baselines (QDT ICML 23, QT ICML 24) to the four neuroevolution tasks we have tested and provide comprehensive comparison results in https://anonymous.4open.science/r/QMamba_review-C0CF/neuroevolution_zero_shot.md. The results there consistently reveal that offline RLs generally underperform online MetaBBO methods, which further underscores the significance of our Q-Mamba. We hope this could address your concern.
>
> **[Novelty]** We would like to argue that the only similarity of our work with Q-transformer is the q-function decomposition scheme. Q-Mamba differentiate with Q-transformer in the following aspects:
>
> 1. **Target tasks**: we have to note that Q-Mamba represents pioneering efforts to explore the possibility of offline RL in MetaBBO tasks, which outperforms online baselines with significantly less training cost . In contrast, Q-transformer, as well as many other offline RLs are developed and examined for classic control tasks.
> 2. **Special designs**: We have explored many customized design choices in this paper to make offline-RL adaptable for MetaBBO:
>     - **Neural network design**: we have combined strength of newly propsoed Mamba architecture into the q- decomposition scheme to boost MetaBBO long sequence tasks (the comparison of Q-transofrmer and Q-Mamba in Table 1 provides a validation).
>     - **Offline dataset construction**: we have proposed a novel dataset construction scheme (Section 4.2) which could collect rigorous exploration and exploitation experiences from diverse baselines. Further ablation study on the data ratio $\mu$ (Section 5.2, Table 3) provide valuable insight of how to make good datasets for offline MetaBBO task.
>     - **Training objective redesign**: compared to the training objective in original CQL and Q-transformer, we additionally add a weight $\beta$ (second case in Eq. (5)) to enhance the q-value learning in the last decomposed action dimensions since the q-value update of the other action dimentions before depends on the accuracy of the last one. The ablation study in Section 5.2 Table 2 demonstrate the effectiveness of such design.
>
> We sincerely request the reviewer to review the above elaboration on the novelty of our work. We appreciate your efforts in reviewing our paper. Please feel free to discuss with us if any concern remains in the author-reviewer discussion period.

---

### Official Review · Reviewer_yyGg · 2025-03-14

**Overall Recommendation:** 2

**Summary:**

The paper introduces Q-Mamba, an offline reinforcement learning framework for Meta-Black-Box Optimization, aimed at efficiently learning Dynamic Algorithm Configuration without online training. It decomposes the Q-function into sequential decisions, applies Conservative Q-Learning to address distribution shift, and uses the Mamba architecture for long-sequence learning. Results show that Q-Mamba performs comparably or better than existing MetaBBO methods.

**Claims And Evidence:**

- Q-Mamba achieves competitive or superior performance to online MetaBBO methods while reducing training costs:
Q-Mamba achieves comparable or slightly superior performance to online MetaBBO baselines such as RLPSO, LDE, and GLEET while reducing training costs. However, since the offline dataset is collected using these same methods, it is expected that Q-Mamba should at least match their performance. In offline RL the goal is to outperform the logged policies used for training. Looking at Table 1, the reported differences in performance metrics between Q-Mamba and the baselines are relatively small, raising the question of how meaningful these improvements are. It is unclear whether these small gains translate into a real advantage.

- Q-function decomposition reduces learning complexity in high-dimensional configuration spaces:
The paper argues that decomposing the Q-function simplifies learning in high-dimensional configuration spaces. However, the experiments use action spaces with only 3, 10, and 16 dimensions, which are not particularly large compared to standard RL environments like Humanoid (17-dimensional) or Ant (8-dimensional). Given this, it is unclear whether decomposition is necessary in these cases.
Additionally, Q-Mamba predicts the action values sequentially, which raises concerns about whether the ordering of action dimensions affects learning and performance. If the sequence matters, it could introduce biases that the paper does not address. A more thorough evaluation would be needed to justify whether decomposition provides a real advantage, particularly in larger action spaces.

**Essential References Not Discussed:**

I think the relevant literature is discussed, but I don’t have extensive knowledge about the field of MetaBBO.

**Experimental Designs Or Analyses:**

I did. Please check previous sections.

**Methods And Evaluation Criteria:**

The proposed methods and evaluation criteria seem to align with the Meta-Black-Box Optimization (MetaBBO) problem; however, I do not have specific expertise or background in MetaBBO to fully assess their appropriateness. The use of offline reinforcement learning makes sense given the inefficiency of online MetaBBO methods, and the CoCo BBOB benchmark suite appears to be a reasonable choice for evaluating optimization performance. The inclusion of both online (RLPSO, LDE, GLEET) and offline baselines (DT, DeMa, QDT, QT, Q-Transformer) provides a comprehensive comparison.

However, one potential concern is that Q-Mamba is trained on data generated by RLPSO, LDE, and GLEET, yet it is later compared against them. The reported improvements are relatively small, raising questions about the significance of these gains.

**Other Comments Or Suggestions:**

None.

**Other Strengths And Weaknesses:**

Strengths:
- The authors share the code.

**Questions For Authors:**

- Please check the previous concerns.
- The conservatism in Equation 5 is not clear. How is it related to CQL? In CQL, out-of-distribution (OOD) actions are sampled, and their estimated Q-values are explicitly decreased to ensure that the learned policy avoids selecting them.
- How many seeds were used?

**Relation To Broader Scientific Literature:**

The paper's contributions relate to prior work in Meta-Black-Box Optimization (MetaBBO), offline reinforcement learning (RL), and sequence modeling. It builds on existing MetaBBO methods like RLPSO and GLEET, by introducing an offline learning framework to improve efficiency. In offline RL, it applies Conservative Q-Learning (CQL) but modifies the regularization term to constrain Q-values of unseen actions. The Q-function decomposition approach aligns with Q-Transformer’s autoregressive Q-learning. Finally, it adopts Mamba, a state-space model, as an alternative to Transformer-based architectures for long-sequence learning, though without direct empirical comparison to validate its advantage.

**Theoretical Claims:**

This work makes no theoretical claims.

---

> ### Author Rebuttal · Authors · 2025-03-31
>
> We appreciate reviewer #yyGg for the thorough review and valuable comments. For the concerns raised above, we provide following point-to-point responses to address them.
>
> **[Performance improvement significance]** We would like to first clarify that **the seemingly small relative performance improvement** of Q-Mamba against the online baselines (RLPSO, LDE and GLEET) it learns from **is mainly due to the normalized metrics** we used for better presentation (as explained in the end of page 6, Section 5.1). Based on the above elaboration, the seemingly small relative performance improvement in Table **actually represents significant absolute performance improvement**. To demonstrate this, we additionally provide the absolute optimization performance comparison results in https://anonymous.4open.science/r/QMamba_review-C0CF/Numerical_results.md, where we provide 3 tables ($Alg0 \sim Alg2$), each for the tested 8 problem instances in CoCo-BBOB testsuites, showing the average best found objective value and error bar across the same 19 independent runs. We hope this could clear the reviewer’s concern. We would add these results into the appendix if the paper was accepted and remind the readers in main text to check them.
>
> **[Decomposition necessity]** We provide comparison results on two additional BBO algorithm sampled from the same ConfigX (https://arxiv.org/abs/2412.07507) modular algorithm space, which hold 22 and 37 hyper-parameters respectively. The offline dataset preparation and the settings follow those in our experiments. Due to the 5000 characters limitation in rebuttal, we provide the results in https://anonymous.4open.science/r/QMamba_review-C0CF/Decomposition_necessity.md where the effectiveness of the decomposition scheme is further validated by the consistent performance superiority of Q-Mamba to both the non-decomposed online methods (RLPSO, LDE and GLEET) and offline methods. We would add these results into the paper if it was accepted and respectifully request the reviewer to check them.
>
> **[Order of action dimension]** We would like to clarify Q-Mamba does not have to address the order bias since we use the pre-order traversal on a legal algorithm structure (represent the algorithm structure as a workflow tree and traversal it in a depth-first way), which can naturally reduce the permutations of the operators in a BBO algorithm to a definitive ordered permutation. On the one hand, doing so could apparently reduce the training difficulty. On the other hand,  the auto-regressive learning in Q-Mamba could implicitly learns the semantic context of the algorithm structure, hence improving the potential generalization performance across various algorithm structures. We thanks the reviewers for this insightful comments and would add this elaboration in our paper to improve the clarity of the methodology.
>
> **[Relation with CQL]** Let us make a quick explanation of the relation between the training objective in Eq. (5) and CQL. First, we adopt the original definition of CQL in Eq. (1) of  its paper (https://arxiv.org/pdf/2006.04779). We also align our training objective with the implemented version in Eq. (2) of the q-Transformer paper (https://arxiv.org/abs/2309.10150). In our decomposition setting, the former two cases in Eq. (5) apply OOD sampling for the policy evaluation. Specifically, we sample maximum q values of the next action step $\max \limits_{j} Q_{i+1,j}^t$ for the action dimension within the same time step yet before the last one, and $\max \limits_{j} Q_{1,j}^{t+1}$ for the last one dimension. Due to the potential distribution shift, such maximum sampling possibly varies with the demonstration we have collected, hence we use the third case in Eq. (5) to regularize the OOD actions, reducing their q-values to the minimal accumulated reward that could be achieved in the optimization process, which is 0 in this case. We respectifully request the reviewer to check the above explanation and hope this could address your concern.
>
> **[Random seeds]** We use three random seeds (1, 333, 9485) for the training of Q-Mamba and other baselines. We use 19 random seeds (100, 200,…,1900) for testing baselines on the CoCo-BBOB problem isntances. We use 10 random seeds (100,200,…,1000) for testing baselines on Neuroevolution problems.

---

### Decision · Program_Chairs · 2025-05-01

**Decision:**

Accept (poster)

**Comment:**

This paper proposes a meta-black-box optimization algorithm based on off policy RL that combines several advancements, namely Q-value decomposition and Mamba. Experiments indicate that the algorithm achieves competitive results with the best on-policy approach with less computation. One weakness is that the experiments do not sufficiently show the value of each component of the algorithm through ablations, which I think can be addressed for the camera ready. Therefore, I would recommend weak accept.